# Physiological and Transcriptome Analysis on Diploid and Polyploid *Populus ussuriensis* Kom. under Salt Stress

**DOI:** 10.3390/ijms23147529

**Published:** 2022-07-07

**Authors:** Hui Zhao, Huanzhen Liu, Jiaojiao Jin, Xiaoyu Ma, Kailong Li

**Affiliations:** 1State Key Laboratory of Tree Genetics and Breeding, Northeast Forestry University, Harbin 150040, China; zhao_hui_zh@163.com (H.Z.); zhenzhen_0522@163.com (H.L.); jinjiaojiao99@163.com (J.J.); mxy12172022@163.com (X.M.); 2College of Forestry, Henan Agricultural University, Zhengzhou 450002, China

**Keywords:** *P. ussuriensis*, triploid, tetraploid, salt stress, response

## Abstract

*Populus ussuriensis* Kom. is a valuable forest regeneration tree species in the eastern mountainous region of Northeast China. It is known that diploid *P. ussuriensis* (CK) performed barely satisfactorily under salt stress, but the salt stress tolerance of polyploid (i.e., triploid (T12) and tetraploid (F20)) *P. ussuriensis* is still unknown. In order to compare the salt stress tolerance and salt stress response mechanism between diploid and polyploid *P. ussuriensis*, phenotypic observation, biological and biochemistry index detections, and transcriptome sequencing (RNA-seq) were performed on CK, T12, and F20. Phenotypic observation and leaf salt injury index analysis indicated CK suffered more severe salt injury than T12 and F20. SOD and POD activity detections indicated the salt stress response capacity of T12 was stronger than that of CK and F20. MDA content, proline content and relative electric conductivity detections indicated CK suffered the most severe cell-membrane damage, and T12 exhibited the strongest osmoprotective capacity under salt stress. Transcriptome analysis indicated the DEGs of CK, T12, and F20 under salt stress were different in category and change trend, and there were abundant *WRKY*, *NAM*, *MYB* and *AP2/ERF* genes among the DEGs in CK, T12, and F20 under salt stress. GO term enrichment indicated the basic growth progresses of CK, and F20 was obviously influenced, while T12 immediately launched more salt stress response processes in 36 h after salt stress. KEGG enrichment indicated the DEGs of CK mainly involved in plant–pathogen interaction, ribosome biogenesis in eukaryotes, protein processing in endoplasmic reticulum, degradation of aromatic compounds, plant hormone signal transduction, photosynthesis, and carbon metabolism pathways. The DEGs of T12 were mainly involved in plant–pathogen interaction, cysteine and methionine metabolism, phagosomes, biosynthesis of amino acids, phenylalanine, tyrosine and tryptophan biosynthesis, plant hormone signal transduction, and starch and sucrose metabolism pathways. The DEGs of F20 were mainly involved in plant hormone signal transduction, plant–pathogen interaction, zeatin biosynthesis, and glutathione metabolism pathways. In conclusion, triploid exhibited stronger salt stress tolerance than tetraploid and diploid *P. ussuriensis* (i.e., T12 > F20 > CK). The differences between the DEGs of CK, T12, and F20 probably are the key clues for discovering the salt stress response signal transduction network in *P. Ussuriensis*.

## 1. Introduction

In the modern world, the growing environment of natural plants is surrounded by various biotic (i.e., herbivores, pathogens, weeds, etc.) and abiotic (i.e., water, salinity, temperature and light) stress factors [1,2,3]. As one of the most common stress factors that plant have to face, salt stress abidingly plays a negative effect role in the whole life cycle of plant [4,5,6]. In order to adapt to a natural salinity environment, different plant species developed different levels of salt stress tolerance in the evolutionary process. Meanwhile, different varieties/subspecies or ploidies of the same species also probably exhibit different levels of salt stress tolerance [7,8]. 

The term “polyploid” refers to the cells and organisms which contain more than two paired (homologous) sets of chromosomes. Additionally, it is reported that polyploidization events perpetually play vital roles in the evolution process of global flora [9,10]. Generally speaking, polyploidy is a common phenomenon in the plant kingdom. For instance, wheat (*Triticum aestivum*), maize (*Zea mays*), banana (*Musa nana* Lour.), oat (*Avena sativa*), strawberry (*Fragaria* × *ananassa* Duch.), Florists Dendranthema (*Dendranthema morifolium* (Ramat.) Tzvel.), alfalfa (*Medicago sativa*), potato (*Solanum tuberosum*), cotton (*Gossypium hirsutum*), and coffee (*Coffea arabica*), etc., are all polyploidy plants [11,12,13,14,15,16,17]. The polyploidy plants include autopolyploid (whose chromosome complement consists of more than two complete copies of the genome of a single ancestral species) and allopolyploid (whose chromosomes are composed of more than two genomes each of which have been derived more or less completely but possibly modified from one of two or more species), and most natural polyploidy plants are allopolyploid [18,19]. Polyploidy may occur due to abnormal cell division (during mitosis or metaphase I in the meiosis phase). Furthermore, polyploidy can also be induced in plants and cell cultures by some chemicals: the best known is colchicine, which can result in chromosome doubling, although its use may have other less obvious consequences as well. According to the previously published report, colchicine-induced octaploid tobacco performed better than tetraploid tobacco under drought- and cold-stress conditions [8]. In addition, tetraploid *Spathiphyllum wallisii* exhibited stronger drought-stress tolerance than its diploid plants [20]. 

As one of the main forest regeneration tree species in the eastern mountainous region of Northeast China, *Populus ussuriensis* Kom. (*P*. *ussuriensis*, belonging to Section *Tacamahaca*) are also distributed in Korea and the Far East Asia region of Russia. The white wood of *P*. *ussuriensis* is light and soft, and it is widely utilized in construction, shipbuilding, papermaking, and matchstick-manufacturing industries [21,22,23]. Therefore, *P*. *ussuriensis* is an important commercial tree species in the forest of Northeast China. According to the previous research by our colleagues, the normal diploid *P*. *ussuriensis* plants were treated with colchicine to produce tetraploid (autopolyploid) *P*. *ussuriensis*, and then the diploid and tetraploid (autopolyploid) *P*. *ussuriensis* were utilized for hybridization to produce triploid *P*. *ussuriensis* [24,25]. 

According to our previous study, diploid *P. ussuriensis* did not exhibit very strong salt stress tolerance [26], while triploid and tetraploid *P. ussuriensis* performed well under drought stress [24]. We are now asking ourselves the question whether triploid and tetraploid *P*. *ussuriensis* also performed unsatisfactorily just like diploid *P. ussuriensis* under salt stress. In consideration of the previously reported results in other polyploid plant species, we have a hypothesis that triploid and tetraploid *P. ussuriensis* probably perform differently to diploid *P. ussuriensis* under salt stress. In order to compare the salt stress tolerance among different ploidies of *P*. *ussuriensis*, the diploid (CK), triploid (T12), and tetraploid (F20) *P*. *ussuriensis* plants were selected as the plant materials for analysis. In the present study, salt stress treatment was taken on CK, T12, and F20 plants; then, the salt-stress-treated functional leaves of CK, T12, and F20 were harvested to perform physiological and biochemical index (superoxide dismutase (SOD) activity, peroxidase (POD) activity, malonaldehyde (MDA) content, proline content, relative electric conductivity, and leaf salt injury index) determinations and transcriptome sequencing (RNA-seq). The results of this study not only can answer the question of whether different ploidies of *P*. *ussuriensis* exhibit different salt stress tolerance, but also can provide valuable clues for the molecular mechanism of salt stress response difference among diploids, triploids and tetraploids of *P*. *ussuriensis*.

## 2. Result

### 2.1. The Leaf Salt Injury Levels of CK, T12, and F20 under Salt Stress

When the plants were cultivated under normal growth conditions (before salt stress treatment), the T12 plants were higher than the CK plants, and the F20 plants were lower than the CK plants, i.e., T12 > CK > F20 (Figure 1a). Meanwhile, the F20 leaves were broader than T12, and the T12 leaves were broader than the CK leaves, i.e., F20 > T12 > CK (Figure 1b). 

Morphological observation of the CK, T12, and F20 plants indicated that three clones all exhibited salt injury symptoms 10 days after salt stress (Figure 1c). A close-up view of the fifth functional leaves of three clones showed the salt injury spots spread over the whole fifth leaf of CK; the fifth functional leaf of T12 did not exhibit significant salt injury; and half of the fifth leaf of F20 contained salt injury spots (Figure 1d). 

Leaf salt injury index analysis indicated there were significant injury level difference among CK, T12, and F20. CK suffered the most severe salt injury (12 d after salt stress, 50.93 ± 3.57%; 15 d after salt stress, 57.81 ± 3.09%); T12 suffered the most slight salt injury (12 d after salt stress, 24.69 ± 2.36%; 15 d after salt stress, 35.10 ± 3.37%); and F20 suffered medium salt injury (12 d after salt stress, 45.37 ± 3.19%; 15 d after salt stress, 50.94 ± 3.54%) (Figure 2). 

### 2.2. SOD and POD Activity of CK, T12, and F20 Leaves under Salt Stress

In order to explore the antioxidant capacity of three clones under salt stress, the leaf SOD and POD activity of them were detected after salt stress treatment. The detection result showed that the SOD activity of CK (39.22 ± 2.83 U g^−1^) was a little lower than T12 (47.69 ± 2.22 U g^−1^) and F20 (48.74 ± 3.67 U g^−1^) under normal growth conditions. Generally speaking, the SOD activity of the three clones first increased and then decreased to normal levels in the 12 days after salt stress treatment. The SOD activity of CK, T12, and F20 reached peak values (CK, 88.80 ± 6.71 U g^−1^; T12, 147.33 ± 9.78 U g^−1^; F20, 117.65 ± 8.14 U g^−1^) on the third day after salt stress (Figure 3a). The POD activity of the three clones in the 12 days after salt stress treatment exhibited a consistent change trend as their SOD activity, i.e., the POD activity of T12 and F20, significantly increased to peak values (T12, 882.14 ± 12.58 U g^−1^; F20, 703.26 ± 16.53 U g^−1^) on the third day after salt stress and then decreased to normal levels, while the POD activity of CK increased to the peak value (CK, 557.24 ± 12.79 U g^−1^) on the sixth day after salt stress and then decreased to a normal level (Figure 3b). 

### 2.3. MDA and Proline Content of CK, T12, and F20 Leaves under Salt Stress

In order to explore the salt injury grade and osmotic regulation capacity of three clones under salt stress, the MDA and proline content of them were detected 0 d, 3 d, 6 d, 9 d, and 12 d after salt stress. The result showed the MDA contents of CK, T12, and F20 leaves have no significant difference under normal growth conditions. Then, the MDA contents of the three clones showed a continuous increasing trend in the 12 days after salt stress, and the CK leaves exhibited the maximum increase rate (267.02%), while the T12 leaves exhibited the minimum increase rate (76.00%) on the third day after salt stress (Figure 4a). In general, proline content of the three clones first increased and then decreased in the 12 days after salt stress. The proline contents of the T12 (23.12 ± 4.69 μg g^−1^) and F20 (22.72 ± 4.19 μg g^−1^) leaves were a little higher than those of the CK (17.51 ± 3.82 μg g^−1^) leaves before salt stress treatment (i.e., normal growth conditions). Then, the proline contents of the three clones increased to peak values (CK, 20.72 ± 4.43 μg g^−1^, increase rate 18.37%; T12, 42.52 ± 4.39 μg g^−1^, increase rate 83.88%; F20, 28.55 ± 3.78 μg g^−1^, increase rate 25.65%) on the third day after salt stress (Figure 4b). 

### 2.4. Relative Electric Conductivity of CK, T12, and F20 Leaves under Salt Stress

In order to evaluate the cell membrane permeability of the three clones, their relative electric conductivities were dynamically detected in the 12 days after salt stress treatment. The result showed the relative electric conductivities of CK, T12, and F20 leaves exhibited a continuous increasing trend in the 12 days after salt stress. Under normal growth conditions (0 d), there were no significant differences between the relative electric conductivity of CK (13.65 ± 0.68%), T12 (13.03 ± 1.47%), and F20 (13.87 ± 1.53%). After salt stress treatment (3 d, 6 d, 9 d, and 12 d), the relative electric conductivities of CK leaves were still significantly higher than that of T12 and F20 leaves, and the T12 leaves still exhibited the minimum relative electric conductivity among the three clones (Table 1). 

### 2.5. Basic Information of RNA-Seq

Finally, 820,030,512 clean reads (min 40,111,090–max 59,470,590 clean reads) were obtained from 18 samples (sample name annotation: T01, T02, and T03 indicate the triple biological repetitions of CK samples under non-salt-stress treatment; T04, T05, and T06 indicate the triple biological repetitions of T12 samples under non-salt-stress treatment; T07, T08, and T09 indicate the triple biological repetitions of F20 samples under non-salt-stress treatment; T10, T11, and T12 indicate the triple biological repetitions of CK samples under salt stress treatment; T13, T14, and T15 indicate the triple biological repetitions of T12 samples under salt stress treatment; and T16, T17, and T18 indicate the triple biological repetitions of F20 samples under salt stress treatment). The percent of clean data quality value (Q) that no less than 30 of each sample ≥91.54%. After alignment, min 70.50%–max 72.80% clean reads per library can be mapped to the reference (*Populus trichocarpa*) genome (Table 2). Based on the alignment result, 284 new genes (189 among them were annotated) were discovered (Appendix A). Additionally, 305,166–351,663 SNPs were discovered and annotated (Appendix A). 

### 2.6. The DEGs among CK, T12, and F20 under Salt Stress Treatment 

Compared with the non-salt-stress-treated CK (T01, T02, T03), T12 (T04, T05, T06), and F20 (T07, T08, T09) samples, there were 2060 (1295 up-regulated DEGs and 765 down-regulated DEGs), 655 (387 up-regulated DEGs and 268 down-regulated DEGs) and 1039 (591 up-regulated DEGs and 448 down-regulated DEGs) DEGs in the salt-stress-treated CK (T10, T11, T12), T12 (T13, T14, T15), and F20 (T16, T17, T18) samples, respectively. A Venn diagram and cluster analysis of the DEGs among T01, T02, T03 vs. T10, T11, T12; T04, T05, T06 vs. T13, T14, T15; and T07, T08, T09 vs. T16, T17, T18 are exhibited in Figure 5 and Appendix A. 

Among the DEGs between T01, T02, T03 vs. T10, T11, T12, there were 31 WRKY transcription factors (TFs), 48 NAM TFs, 45 MYB TFs, and 27 AP2/EREBP TFs. There were 15 WRKY TFs, 11 NAM TFs, 11 MYB TFs, and 17 AP2/EREBP TFs in the DEGs between T04, T05, T06 vs. T13, T14, T15. As for the tetraploid samples, there were 15 WRKY TFs, 16 NAM TFs, 21 MYB TFs, and 27 AP2/EREBP TFs in the DEGs between T07, T08, T09 vs. T16, T17, T18 (Appendix A). 

### 2.7. GO Classification and GO Term Enrichment on the DEGs among CK, T12, and F20 under Salt Stress Treatment

GO classification and GO term (biological process) enrichment on the DEGs between T01, T02, T03 and T10, T11, T12 indicate that most of the significant differentially expressed genes in CK under salt stress were involved in the metabolic process, cellular process, biological regulation, signaling, immune system, and cell-killing processes (Figure 6a and Appendix A). GO classification and GO term (biological process) enrichment on the DEGs between T04, T05, T06 and T13, T14, T15 indicate that most of the significant differentially expressed genes in T12 under salt stress were involved in the cellular process, response to stimulus, signaling and immune system processes (Figure 6b and Appendix A). GO classification and GO term (biological process) enrichment on the DEGs between T07, T08, T09 and T16, T17, T18 indicate that most of the significant differentially expressed genes in F20 under salt stress were involved in growth, developmental, metabolic, signaling, immune system and cell-killing processes (Figure 6c and Appendix A).

### 2.8. KEGG Enrichment on the DEGs among CK, T12, and F20 under Salt Stress Treatment

According to the KEGG enrichment analysis, the DEGs between T01, T02, T03 and T10, T11, T12 were mainly involved in plant–pathogen interaction, ribosome biogenesis in eukaryotes, protein processing in endoplasmic reticulum, degradation of aromatic compounds, plant hormone signal transduction, photosynthesis, and carbon metabolism pathways (Figure 7a). Most of the DEGs between T04, T05, T06 and T13, T14, T15 were involved in plant–pathogen interaction, selenocompound metabolism, cysteine and methionine metabolism, phagosomes, biosynthesis of amino acids, phenylalanine, tyrosine and tryptophan biosynthesis, plant hormone signal transduction, and starch and sucrose metabolism pathways (Figure 7b). Most of the DEGs between T07, T08, T09 and T16, T17, T18 were involved in plant hormone signal transduction, plant–pathogen interaction, zeatin biosynthesis, and glutathione metabolism pathways (Figure 7c).

### 2.9. QRT-PCR Verification

According to the RNA-seq result, *WRKY40* (Potri.001G044500) was significantly up-regulated (37.01 folds), while *WRKY4* (Potri.008G091900) was significantly down-regulated (0.38 folds) in CK under salt stress; *NAM* (Potri.005G069500) was significantly up-regulated (20.39 folds), while *MYB* (Potri.005G001600) and *NAC* (Potri.007G135300) was significantly down-regulated (0.23 and 0.30 folds, respectively) in T12 under salt stress; and *AP2_ERF* (Potri.018G038100) was significantly up-regulated (46.53 folds) in F20 under salt stress. Subsequent qRT-PCR verification indicated that the expression of Potri.001G044500, Potri.005G069500, and Potri.018G038100 was significantly up-regulated, while the expression of Potri.008G091900, Potri.005G001600, and Potri.007G135300 was significantly down-regulated 36 h after salt stress (Figure 8). 

## 3. Discussion

In plant organisms, the superoxide dismutase (SOD) can protect cells against oxidative damages by scavenging the reactive oxygen species which were caused by salinity stress [27,28]. As heme-containing monomeric glycoproteins in plants, peroxidases (POD) catalyze the zymolyte oxidation by utilizing hydrogen peroxide as the electron acceptor. Additionally, as the result of catalysis, hydrogen peroxide and phenols/amines toxicity in plant cells are effectively obviated [29]. In the present study, the SOD and POD activity of diploid was a little lower than triploid and tetraploid *P. ussuriensis* under normal growth conditions, and this phenomenon is probably due to their genome size difference [30,31]. The SOD activity of CK, T12, and F20 significantly increased to peak values in three days and then decreased to normal levels after salt stress, and the SOD activity increase rate of T12 (208.91%) on the third day was significantly higher than CK (126.43%) and F20 (141.37%). The POD activity of T12 and F20 significantly increased to peak values (increase rate, T12, 123.27%; F20, 83.17%) in three days and then decreased to normal levels, while the POD activity of CK reached the peak value (increase rate 53.17%) on the sixth day and then decreased to a normal level after salt stress. These results indicate the salt stress response capacity of T12 was stronger than that of CK and F20 (i.e., T12 > F20 > CK). 

Malondialdehyde (MDA) is one of the final products of peroxidation of unsaturated fatty acids in phospholipids, and its content can reflect lipid peroxidation or cell-membrane damage in plant tissues [32,33]. Proline plays a typical osmoprotectant role when the plant suffered osmotic stress (such as salinity and drought), and the instant proline accumulation presents osmoprotective capacity [34,35,36]. Relative electric conductivity is a well-recognized physiological index which can reflect the osmotic regulation capacity of the plasma membrane under salt stress [37]. In this study, the MDA contents of three clones all exhibited a continuously increasing trend in the 12 days after salt stress, and the increase rate of T12 was significantly higher than CK and F20. The proline contents of three clones increased to peak values on the third day after salt stress, and the increase rate of T12 was the maximum. Relative electric conductivity of CK leaves was still significantly higher than that of T12 and F20 leaves, and the T12 leaves still exhibited the minimum relative electric conductivity among the three clones in the 12 days after salt stress. These results indicate CK suffered the most severe cell-membrane damage, and T12 exhibited the strongest osmoprotective capacity, among the three clones, under salt stress.

According to phenotypic observation and leaf salt injury index analysis, CK leaves suffered the most severe (i.e., CK > F20 > T12) salt injury among three clones in the 15 days after salt stress. Combined with the above physiology and biochemistry indexes (SOD and POD activity, MDA and proline content, relative electric conductivity) detection results, we can conclude that the triploid *P. ussuriensis* exhibited the strongest salt stress tolerance, while diploid *P. ussuriensis* exhibited sensitivity to salinity. 

WRKY, NAM, MYB, and AP2/EREBP TFs were all fatal regulation nodes in salt stress response signal transduction processes in plant [38,39,40,41,42]. Generally speaking, there were abundant WRKY, NAM, MYB, and AP2/ERF family members clustered among the DEGs in CK, T12, and F20 under salt stress. However, the detail change fold and amount of these differentially expressed *WRKY*, *NAM*, *MYB,* and *AP2/ERF* genes were not identical in CK, T12, and F20. For instance, *WRKY3* (Potri.008G091900) was down-regulated in CK, while it was not differentially expressed in T12 and F20 under salt stress. *WRKY40* (Potri.001G044500) was significantly up-regulated, and *MYB86* (Potri.005G001600) was significantly down-regulated in CK, T12, and F20 under salt stress. *MYB39* (Potri.004G033100) was up-regulated in CK, while it was not differentially expressed in T12 and F20 under salt stress. These results indicate that there were not only similar salt-stress-responsive mechanisms but also different salt-stress-response signal transduction mechanisms in diploid, triploid, and tetraploid *P. ussuriensis*. Exploring the biological function of these special DEGs (such as Potri.008G091900 and Potri.004G033100) is valuable for the further refinement of the salt-stress-response signal transduction network of *P. ussuriensis*. 

GO term enrichment indicated that most of the DEGs of T12 are involved in response to stimulus, signaling, and immune system processes. As for the DEGs of CK and F20, except for some stress response processes, an amount of the DEGs of CK and F20 are involved with the metabolic process, cellular process, growth, and developmental progress. These phenomena indicated that the basic growth processes of CK and F20 were obviously influenced in 36 h under salinity stress, while T12 immediately launched salt stimulus response processes. 

KEGG enrichment indicated the DEGs of CK were mainly involved in plant–pathogen interaction, ribosome biogenesis in eukaryotes, protein processing in endoplasmic reticulum, degradation of aromatic compounds, plant hormone signal transduction, photosynthesis, and carbon metabolism pathways. Endoplasmic reticulum (ER) stress can induce the accumulation of unfolded protein in ER and further induce apoptosis [43,44]. Plant aromatic compounds such as phenolics are antioxidants under oxidative stress [45]. Plant hormone signal transduction is also a common stress response pathway [46,47]. Meanwhile, photosynthesis and carbon metabolism are basic growth- and development-related pathways of plants. The DEGs of T12 were mainly involved in plant–pathogen interaction, cysteine and methionine metabolism, phagosomes, biosynthesis of amino acids, phenylalanine, tyrosine and tryptophan biosynthesis, plant hormone signal transduction, and starch and sucrose metabolism pathways. Cysteine is the major component of glutathione, while glutathione is related to the plant abiotic-stress response. Methionine can control the production of stress metabolites such as ethylene and glucosinolates. Therefore, cysteine and methionine metabolism are important abiotic-stress-response processes [48,49]. Glucosinolates are known as plant-defense-reaction-related compounds. Indole and aromatic glucosinolates are derived from tryptophan and phenylalanine or tyrosine. Therefore, the biosynthesis of phenylalanine, tyrosine, and tryptophan is also a vital stress response process in plants [50]. The DEGs of F20 were mainly involved with plant hormone signal transduction, plant–pathogen interaction, zeatin biosynthesis, and glutathione metabolism pathways. Zeatin is an important hormone in plant biotic- and abiotic-stress responses [51,52]. Glutathione also plays an important role in plant abiotic stress tolerance [53,54]. In addition, the enriched pathway, plant–pathogen interaction, indicated CK, T12, and F20 probably meet with pathogen stress from the air. 

In conclusion, the present study analyzed and compared the salt stress tolerance of diploid, triploid, and tetraploid *P. ussuriensis* on phenotypic, physiological, biochemical, and transcriptome levels. The results demonstrated that our hypothesis is correct, i.e., triploid and tetraploid *P. ussuriensis* indeed displayed different salt stress tolerance with diploid *P. ussuriensis*. Phenotypic, physiological, and biochemical (SOD and POD activity, MDA and proline content, relative electric conductivity) differences between CK, T12, and F20 under salt stress indicated diploid, triploid, and tetraploid *P. ussuriensis* launched different molecular-response mechanisms to deal with salt stress. Vice versa, the RNA-seq results (such as the above-mentioned DEG differences between CK, T12, and F20, *WRKY3* (Potri.008G091900), *MYB86* (Potri.005G001600) and *MYB39* (Potri.004G033100), etc.) probably can explain the phenotypic, physiological, and biochemical differences among CK, T12, and F20 under salt stress on a molecular level. In further research, the biological functions and salt-stress-response-signaling mechanisms of these special DEGs (*WRKY3*, *MYB86,* and *MYB39*) should be studied and verified through a transgenic system, yeast one- and yeast two-hybrid assays, chromatin immunoprecipitation, and other biological or biochemical experiments. 

## 4. Materials and Methods

### 4.1. Plant Materials and Salt Stress Treatment

The diploid (CK), triploid (T12), and tetraploid (F20) of *P. ussuriensis* plants were first grown in a substrate (1/2 Murashige and Skoog (1/2 MS) basal medium (PhytoTechnology Laboratories, Shawnee Mission, KS, USA) supplemented with 0.1 mg L^−1^ 1-naphthaleneacetic acid (NAA) (Sigma-Aldrich, St. Luis, MO, USA), and then were transplanted into plastic pots (20 cm diameter × 18 cm depth) that contained sand and peat (*v*/*v* = 1:2). The transplanted plants were cultured in a culture room of State Key Laboratory of Tree Genetics and Breeding, Northeast Forestry University (Harbin, Heilongjiang, China, northern latitude 45°43′, eastern longitude 126°38′) under the following growth conditions: culture temperatures, 20 °C–28 °C; relative humidity, 55 ± 5%; photoperiod, 16 h light/8 h dark; photosynthetic photon flux, 150 μmol m^−2^ s^−1^. When all the plants grew to approximately 30 cm, they were subjected to salt stress (120 mM NaCl) treatment. The NaCl solution was supplied to CK, T12, and F20 plants for 24 h. For physiological and biochemical index detection: the fourth to sixth functional leaves of CK, T12, and F20 plants were harvested just before salt stress treatment (i.e., 0 d samples) and 3 d, 6 d, 9 d, and 12 d after salt stress treatment at 8:30 am. Meanwhile, the third functional leaves of CK, T12, and F20 plants (three biological replicates) were collected just before (i.e., 0 h samples) and 36 h after salt stress for the subsequent transcriptome sequencing and qRT-PCR (relative quantitative real-time PCR) verification. After sample collection, the tissues were frozen immediately in liquid nitrogen and stored in −80 °C freezers prior to the further experiments. 

### 4.2. Leaf Salt Injury Index Survey

Twelve and fifteen days after salt stress treatment, the leaf salt injury degrees of CK, T2 and F20 were evaluated (ten biological replicates) by “Leaf salt injury index”. The evaluation standard of “Leaf salt injury index” referenced the computational formula that Zhao et al. utilized [55]: $$\mathrm{Plant}\;\mathrm{leaf}\;\mathrm{salt}\;\mathrm{injury}\;\mathrm{index}=\frac{\sum^{\;}\left(i\;\times {\;N}_{i}\right)}{\left(N_{T}\times {\;G}_{\max}\right)}\times 100\%,$$
where i indicates salt injury level (I = 0−4); N_i_ indicates the number of “i level” leaves; N_T_ indicates the total number of leaves; G_max_ indicates the highest salt injury level (i.e., 4).

### 4.3. SOD and POD Activity, and MDA and Proline Content Detection

Leaf SOD and POD activities, and MDA and proline contents of each line (CK, T12, and F20) were determined by Superoxide Dismutase Assay kit-Visible Spectrophotometry (SOD-2-Y, Comin, Suzhou, China), Peroxidase Assay kit-Visible Spectrophotometry (POD-2-Y, Comin, Suzhou, China), Malonaldehyde Assay kit-Visible Spectrophotometry (MDA-2-Y, Comin, Suzhou, China), and Proline Content Detection kit-Visible Spectrophotometry (PRO-2-Y, Comin, Suzhou, China), respectively, and the detailed detection procedures were conducted by utilizing a Lambda 25 UV/VIS Spectrophotometer (Perkin Elmer, IL, USA) according to the manual that the manufacturers provided. All the samples were detected by three independent repeats. 

### 4.4. Relative Electric Conductivity Detection 

To detect the relative electrolytic leakages of CK, T12, and F20 leaves, approximately 0.3 g leaf tissues were taken from each leaf sample by a puncher. First, the leaf tissues were soaked in distilled water, and their conductivity values (designed as C1) were detected by a conductometer (FiveEasyPlus FE38, METTLER TOLEDO, Zurich, Switzerland). Then, the leaf samples were transferred into boiling water for 20 min, cooled down to room temperature (distilled water was added to make a total volume of 30 mL), and utilized for total conductivity value (designed as C2) detection. All the leaf samples were detected by three independent repeats. The relative electrolytic leakages of all samples were calculated as follows: Relative electrolyte leakage=(C1C2)×100%
where C1 is the first detected conductivity value of each leaf sample, C2 is the total conductivity value of each leaf sample. 

### 4.5. RNA-Seq

In order to further explore the difference between CK, T12, and F20 in molecular level, RNA-seq was performed. The EASYspin Plus Plant RNA kit (Aidlab, Beijing, China) was utilized to extract total RNA from each sample. Then, the total RNA of each sample was taken to perform transcriptome sequencing in Biomarker Technologies Co., Ltd. (Beijing, China): The mRNA was purified by magnetic oligo-dT beads, and was broken with fragmentation buffer. The cDNA was synthesized by the addition of dNTPs, Rnase H, DNA polymerase I, and GEX second strand buffer. The synthesized cDNA was purified with AMPure XP beads (Beckman Coulter, Fullerton, CA, USA), and subsequently eluted in EB buffer for end-repair and poly-A addition. Suitable fragments were selected for PCR amplification by Beckman AMPure XP beads. The PCR amplicons produced a cDNA library with inserts (approximately 200–500 bp in length). Finally, the amplified library was sequenced using high-throughput Illumina HiSeq 4000 platform (Illumina, San Diego, CA, USA), with both ends of the inserts being sequenced. 

Clean reads were obtained by removing artificial linkers, adaptors, and low-quality reads (i.e., <20 bp or reads containing more than 5% unknown nucleotides). High-quality clean reads were then mapped according to the *P*. *trichocarpa* mRNA reference sequence (https://www.ncbi.nlm.nih.gov/genome/?term=Populus+trichocarpa, accessed on 5 December 2021) using the spliced mapping algorithm in TopHat (ver. 2.0.9) [56]. GATK2 software was utilized to perform SNP (single-nucleotide polymorphisms) calling. Raw vcf. Files were filtered with GATK standard filter method and other parameters (clusterWindowSize: 10; MQ0 ≥ 4 and (MQ0/(1.0 × DP)) > 0.1; QUAL < 10; QUAL < 30.0 or QD < 5.0 or Hrun > 5), and only SNPs with distance >5 were retained [57]. Transcript abundance was quantified using the Cufflink module in the Cufflinks program (ver. 2.1.1) [58]. Gene expression levels were measured as fragments per kilobase of transcript per million mapped reads (FPKM). Differentially expressed genes (DEGs) were identified with a false discovery rate (FDR) < 0.05 (the *p* value was adjusted using the Benjamini–Hochberg method [59]) and fold change (FC) ≥ 2. The DEGs and their encoded amino acid sequences were annotated as He et al. described [60]. To identify gene ontology (GO) terms, the DEGs were imported into the Blast2GO program [61]. We completed singular enrichment analyses using the agriGO database (http://bioinfo.cau.edu.cn/agriGO/analysis.php, accessed on 1 December 2021) to identify significantly enriched GO terms among the DEGs. The KEGG Orthology-Based Annotation System 2.0 (KOBAS, http://kobas.cbi.pku.edu.cn/program.run.do, accessed on 1 February 2022) was utilized to identify significantly enriched KEGG pathways [62,63]. 

### 4.6. QRT-PCR

In order to verify the accuracy of RNA-seq, qRT-PCR was performed subsequently. Total RNA was extracted from stress-treated samples using the Universal Plant Total RNA Extraction kit (Spin-column) (type 1) (BioTeke Corporation, Beijing, China). Double-stranded cDNA was synthesized by reverse transcription using the ReverTra Ace qPCR RT Master Mix with gDNA Remover (Toyobo, Osaka, Japan). 

Gene-specific primers for *WRKY40* (Potri.001G044500), *NAM* (Potri.005G069500), *AP2_ERF* (Potri.018G038100), *WRKY4* (Potri.008G091900), *MYB* (Potri.005G001600), *NAC* (Potri.007G135300) (Appendix A) were designed using Primer Premier 6.0 software and the Primer-BLAST tool in NCBI (http://www.ncbi.nlm.nih.gov/tools/primer-blast/index.cgi?LINK_LOC=BlastHome, accessed on 9 June 2022). According to our previous research [64], the housekeeping gene *At4g33380-like* was selected as the reference gene. The qRT-PCR was completed in triplicate using the SYBR Green Real-time PCR Master Mix Plus (Toyobo, Osaka, Japan) and the ABI PRISM 7500 Real-Time PCR system (Applied Biosystems, Foster City, CA, USA). The qRT-PCR program was as follows: 95 °C for 1 min; 40 cycles of 95 °C for 15 s; and 60 °C for 1 min. Dissociation curves were constructed using the thermal-melting profile generated after the last PCR cycle under the following conditions: 95 °C for 15 s followed by a constant increase in temperature from 60 to 95 °C. The final expression data were analyzed using the 2^−ΔΔCt^ method [65].

## 5. Conclusions

Phenotypic observation and leaf salt injury index calculation indicated CK suffered more severe salt injury than T12 and F20 under salt stress. SOD and POD activity detections, MDA content, proline content, and relative electric conductivity detections indicated triploid (T12) *P. ussuriensis* exhibited stronger salt stress tolerance than tetraploid (F20) and diploid (CK) *P. ussuriensis* (i.e., T12 > F20 > CK). 

Transcriptome analysis indicated the DEGs of CK, T12, and F20 under salt stress were different in category and change trend, and there were abundant *WRKY*, *NAM*, *MYB,* and *AP2/ERF* genes among the DEGs in CK, T12, and F20 under salt stress. Additionally, these results indicated that there were not only similarities but also differences (such as *WRKY3* (Potri.008G091900), *MYB86* (Potri.005G001600), and *MYB39* (Potri.004G033100)) between the DEGs of CK, T12, and F20 before and after salt stress. We suspect that these differences are the key clues for discovering the salt stress response mechanism differences between diploid and polyploid *P. ussuriensis*. Therefore, further research should pay more attention to these special DEGs. For instance, the biological functions and salt-stress-response-signaling mechanisms of these special DEGs should be studied and verified through a transgenic system, yeast one- and yeast two-hybrid assays, chromatin immunoprecipitation, and other biological or biochemical experiments. 

## Figures and Tables

**Figure 1 ijms-23-07529-f001:**
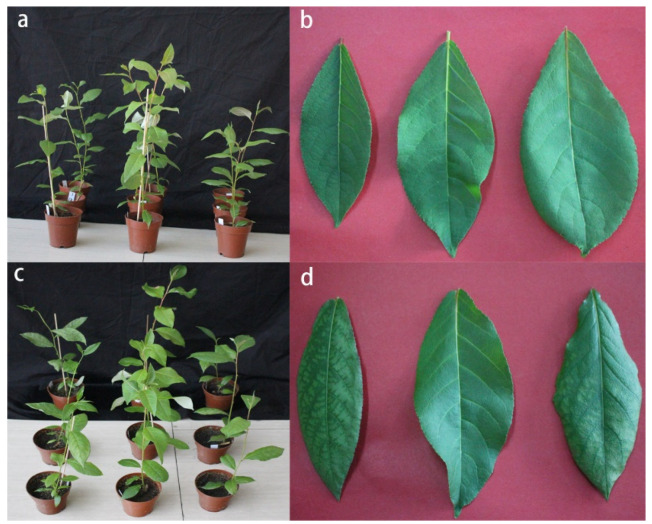
Morphological observation on CK, T12, and F20 plants and functional leaves before and 10 days after salt stress. (**a**) The plants from left to right indicate CK, T12, and F20 plants under normal growth conditions (i.e., before salt stress). (**b**) The leaves from left to right indicate the fifth functional leaves of CK, T12, and F20 under normal growth conditions (i.e., before salt stress). (**c**) The plants from left to right indicate CK, T12, and F20 plants 10 days after salt stress. (**d**) The leaves from left to right indicate the fifth functional leaves of CK, T12, and F20 10 days after salt stress.

**Figure 2 ijms-23-07529-f002:**
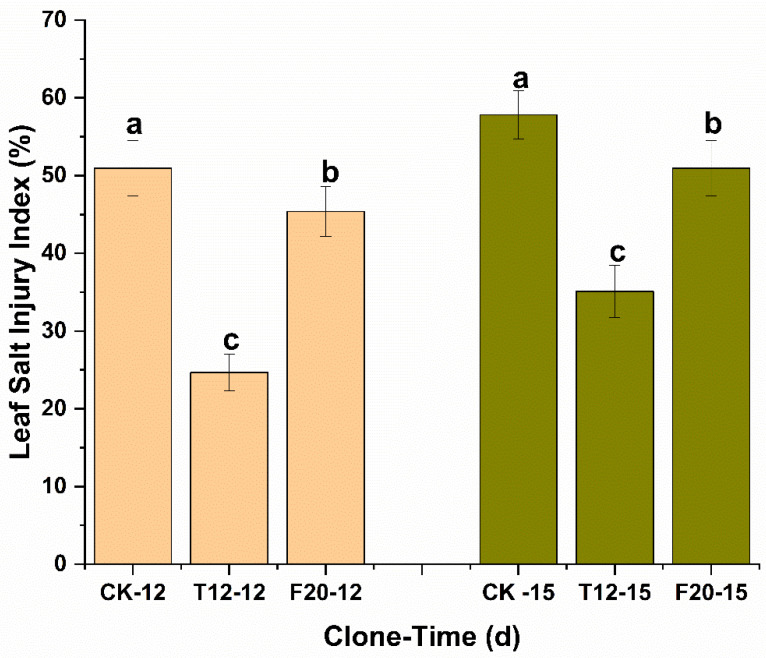
Leaf salt injury index of CK, T12, and F20. CK-12, T12-12, and F20-12 indicate the leaf salt injury index of CK, T12, and F20 on the 12th day after salt stress treatment; CK-15, T12-15, and F20-15 indicate the leaf salt injury index of CK, T12, and F20 on the 15th day after salt stress treatment.

**Figure 3 ijms-23-07529-f003:**
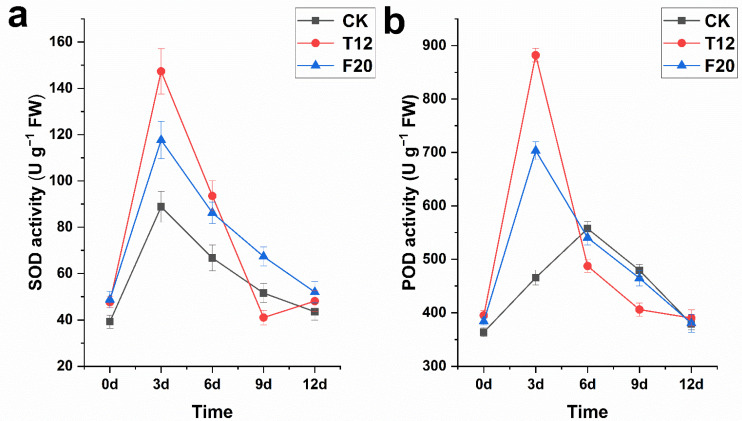
SOD and POD activity of CK, T12, and F20 before and under salt stress.

**Figure 4 ijms-23-07529-f004:**
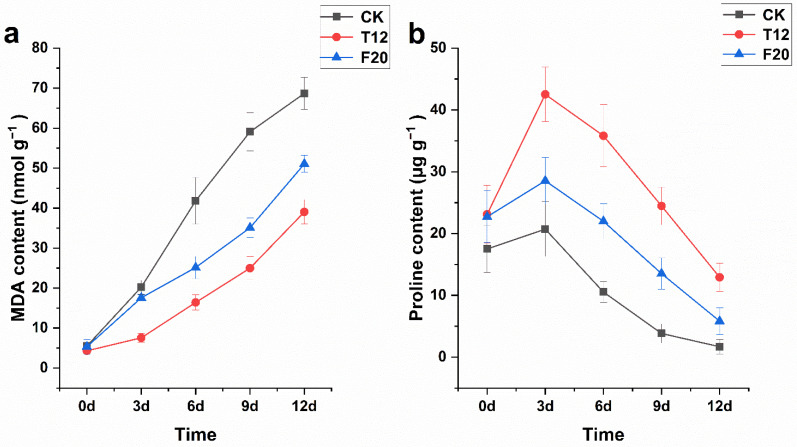
MDA and Proline content of CK, T12, and F20 before and under salt stress.

**Figure 5 ijms-23-07529-f005:**
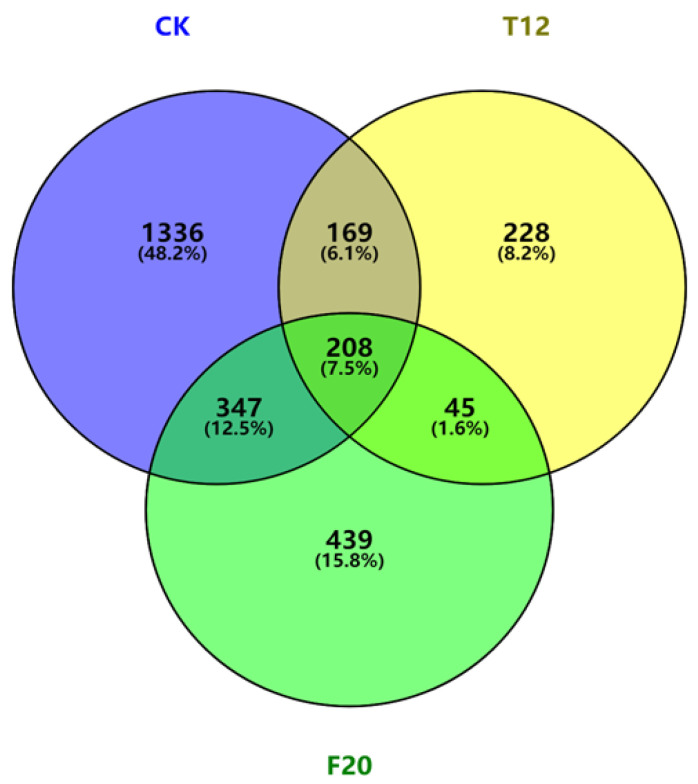
Venn diagram of the DEGs among CK, T12, and F20 under salt stress treatment. CK indicates the DEGs between T01, T02, T03 and T10, T11, T12 samples; T12 indicates the DEGs between T04, T05, T06 and T13, T14, T15 samples; F20 indicates the DEGs between T07, T08, T09 and T16, T17, T18 samples.

**Figure 6 ijms-23-07529-f006:**
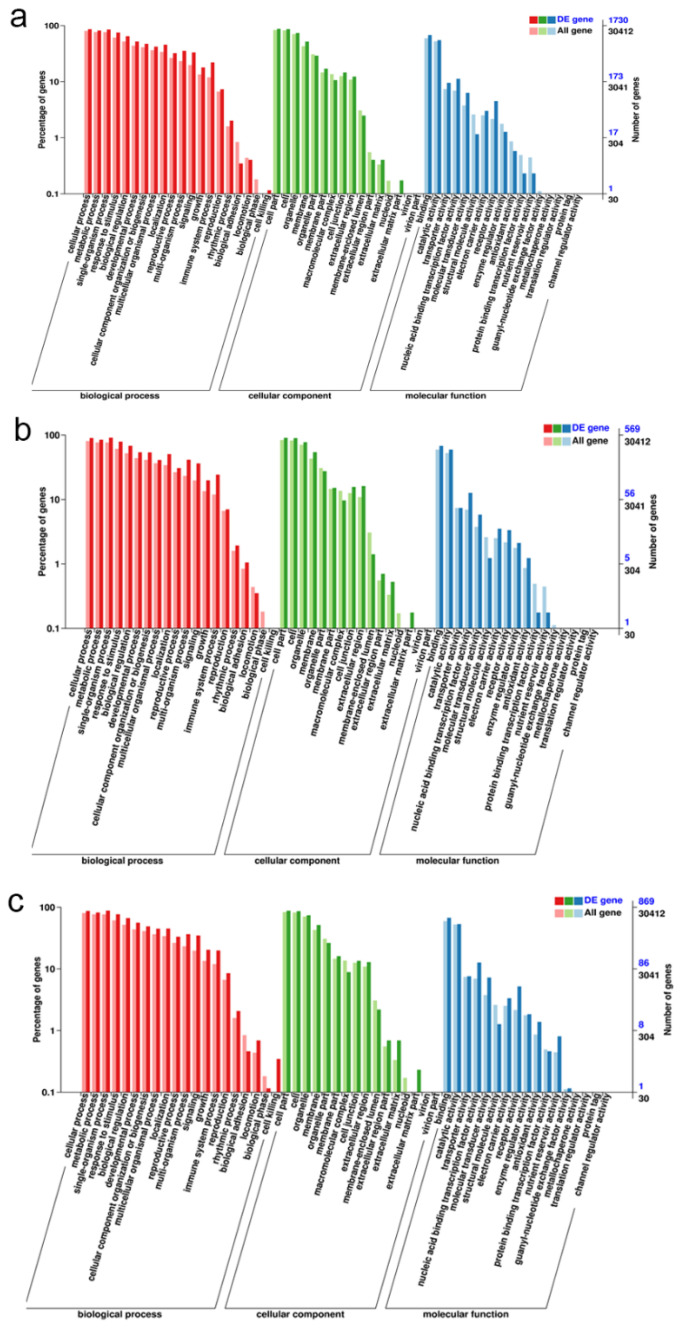
GO classification on the DEGs among CK, T12, and F20 under salt stress treatment. (**a**) GO classification of the DEGs between T01, T02, T03 and T10, T11, T12 samples; (**b**) GO classification of the DEGs between T04, T05, T06 and T13, T14, T15 samples; (**c**) GO classification of the DEGs between T07, T08, T09 and T16, T17, T18 samples.

**Figure 7 ijms-23-07529-f007:**
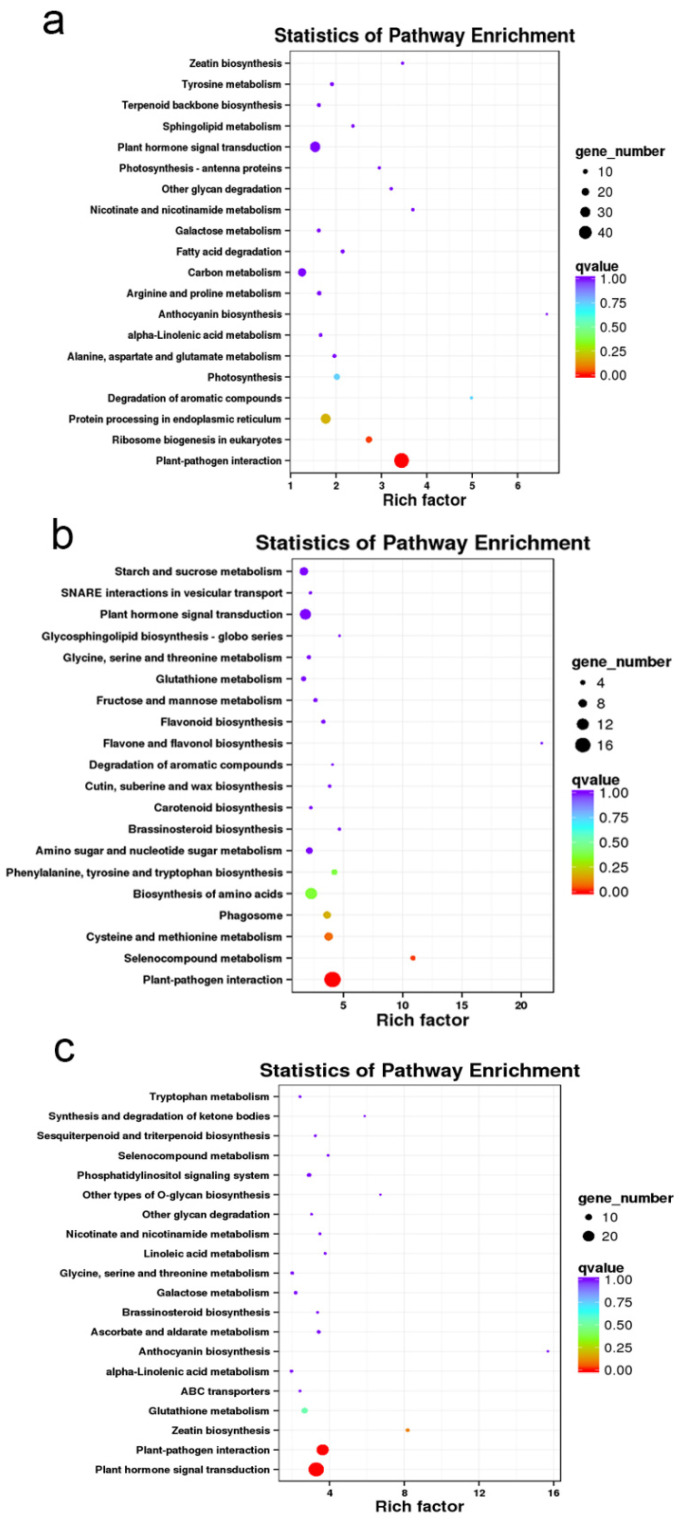
KEGG enrichment on the DEGs among CK, T12, and F20 under salt stress treatment. (**a**) KEGG enrichment of the DEGs between T01, T02, T03 and T10, T11, T12 samples; (**b**) KEGG enrichment of the DEGs between T04, T05, T06 and T13, T14, T15 samples; (**c**) KEGG enrichment of the DEGs between T07, T08, T09 and T16, T17, T18 samples.

**Figure 8 ijms-23-07529-f008:**
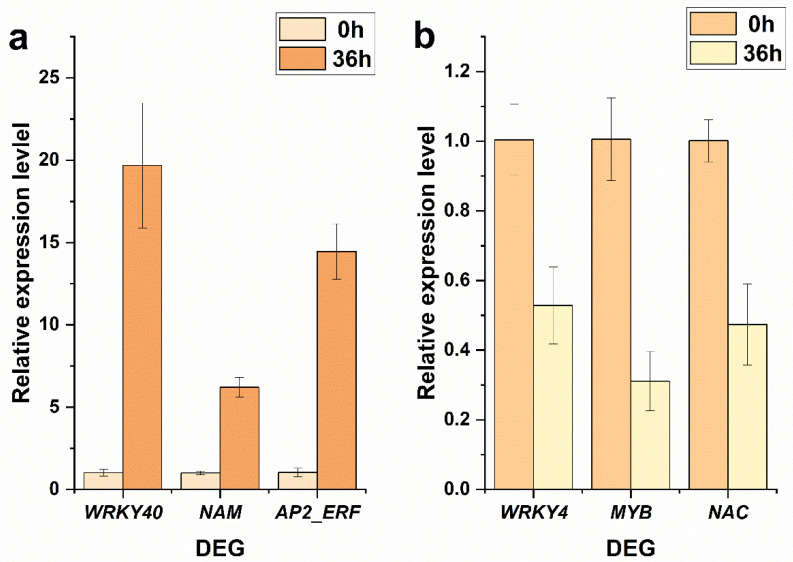
QRT-PCR verification on the expression change trends in several DEGs before and after salt stress. (**a**) QRT-PCR verification on the expression change trends of three up-regulated DEGs (*WRKY40*, Potri.001G044500; *NAM*, Potri.005G069500; *AP2_ERF*, Potri.018G038100). (**b**) QRT-PCR verification on the expression change trends of three down-regulated DEGs (*WRKY4*, Potri.008G091900; *MYB*, Potri.005G001600; *NAC*, Potri.007G135300).

**Table 1 ijms-23-07529-t001:** The relative electrical conductivities of CK, T12, and F20 leaves.

Clone	REC-0 d	REC-3 d	REC-6 d	REC-9 d	REC-12 d
CK	17.65 ± 1.16 a	21.09 ± 1.84 a	25.96 ± 1.35 a	31.02 ± 3.00 a	35.64 ± 0.85 a
T12	11.70 ± 1.37 b	15.19 ± 1.44 b	16.45 ± 1.03 c	19.03 ± 1.40 c	22.79 ± 1.59 c
F20	13.87 ± 1.62 b	16.92 ± 0.19 b	19.37 ± 1.08 b	24.36 ± 1.02 b	28.69 ± 1.42 b

REC indicates the relative electrical conductivity, %. Data indicate means ± STDEV, “a, b, c “indicate the significance of difference (n = 3, *p* < 0.05, multiple-comparison method: Duncan).

**Table 2 ijms-23-07529-t002:** Basic information of RNA-seq of CK, T12, and F20 before and after salt stress.

Sample Name	Clean Reads	Mapped Clean Reads	%≥ Q30
T01	41,910,366	30,375,971	92.38%
T02	41,040,924	29,443,067	92.26%
T03	40,111,090	28,728,047	92.32%
T04	44,212,562	32,116,907	91.93%
T05	45,811,970	32,832,194	91.54%
T06	42,112,362	30,218,367	92.16%
T07	41,468,618	30,189,953	91.86%
T08	45,156,026	32,502,831	92.58%
T09	40,742,788	28,892,712	91.59%
T10	40,754,178	28,732,594	91.63%
T11	45,714,746	32,271,613	91.74%
T12	45,579,522	32,301,725	92.16%
T13	57,068,718	40,927,502	92.16%
T14	45,700,546	32,576,326	91.78%
T15	47,126,662	33,622,560	92.31%
T16	48,398,450	34,772,614	92.76%
T17	59,470,590	42,382,740	92.82%
T18	47,650,394	34,081,348	92.47%

Sample name annotation: T01, T02, and T03 indicate the triple biological repetitions of diploid samples under non-salt-stress treatment; T04, T05, and T06 indicate the triple biological repetitions of triploid samples under non-salt-stress treatment; T07, T08, and T09 indicate the triple biological repetitions of tetraploid samples under non-salt-stress treatment; T10, T11, and T12 indicate the triple biological repetitions of diploid samples under salt stress treatment; T13, T14, and T15 indicate the triple biological repetitions of triploid samples under salt stress treatment; T16, T17, and T18 indicate the triple biological repetitions of tetraploid samples under salt stress treatment. Mapped Clean Reads, the amount of clean reads which were mapped to the reference (*P. trichocarpa*) genome; %≥ Q30, the percentages of the bases whose Quality Score ≥ 30 (the base calling error rate ≤ 0.1%).

## Data Availability

Not applicable.

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
