# Peer review of "Physiological and Transcriptome Analysis on Diploid and Polyploid Populus ussuriensis Kom. under Salt Stress"

_ijms, 2022, doi:10.3390/ijms23147529_

Round 1

Reviewer 1 Report

After water stress, salinity stress is one of the most important abiotic stressors that affect plants. The study of the effect of salinity on plants is therefore still relevant, as plants differ not only within plant species but also between genotypes. The manuscript deals with the study of the influence of polyploidy on the resistance of poplar plants to salinity. The manuscript is written relatively carefully. In the methodology, I miss the method of growing plants. The authors report only the concentration of NaCl. Were the plants grown in a substrate or hydroponically? What was the chemical composition of the substrate? How often was the NaCl solution supplied to the plants? The results are relatively well presented. It is necessary to modify figures 6 and 7, for which the description of the axes is illegible. It would be appropriate to add correlations between the obtained results to the results. Complement the discussion based on correlation relations. Add and edit summary. Journales are inconsistently cited in the literature review. Needs to be adjusted.

Author Response

First of all, we would like to express our appreciation for your valuable advices.

  • The CK, T12 and F20 plants were first grown in a substrate (1/2 Murashige and Skoog (1/2 MS) basal medium (PhytoTechnology Laboratories, Shawnee Mission, KS, USA) supplemented with 0.1 mg l−1 1-naphthaleneacetic acid (NAA) (Sigma-Aldrich, St. Luis, MO, USA)), and then were transplanted into plastic pots (20 cm diameter × 18 cm depth) that containing sand and peat (v/v = 1:2). The NaCl solution was supplied to CK, T12 and F20 plants per 24h. And the complement information has been added in the Materials and methods of the marked revised manuscript. (Line 98-109)
  • We are sorry that the quality of Figure 6 and Figure 7 was not very well displayed in the docx. manuscript. The original files of Figure 6 and Figure 7 have been uploaded to the submission system, and the distinguishability of the two .tif files were both 600 dpi.
  • The correlations between the obtained phenotypic, physiological and biochemical, and transcriptome results in this study have been added in the Discussion of the marked revised manuscript. (Line 460-475)
  • The summary (Conclusions) has been added after the Discussion of the marked revised manuscript. (Line 478-498)
  • The references have been adjusted according to your advice in the marked revised manuscript. (Line 539-671)

Reviewer 2 Report

Dear Authors,

Work entitled: "Physiological and transcriptomic analysis of diploids and polyploids Populus ussuriensis Com. under the influence of salt stress "is interesting, both from a scientific and practical point of view, and deserves to be published in the journal "IJMS". However, in order to raise work to a higher level, several conditions must be met:

1.     Structure the Abstract: background, research aim, short methodology, Results and conclusion.

2.     At the end of the introduction, an alternative hypothesis should be put forward to the null hypothesis and verified later in the paper.

3.     There is no subsection on statistical calculations in the research methodology, it should be supplemented.

4.     In table 1, the letters should be at the mean value and not at the value of the standard deviation in the legend of table, it should be explained what they mean.

5.     The discussion is quite short and does not cover all aspects of the research. First of all, it is necessary to verify the alternative research hypothesis (what was assumed before the experiment and whether these assumptions were confirmed).

6.     There should be a chapter called Conclusions, and one chapter should indicate the utilitarian nature of the research

Author Response

First of all, we would like to express our appreciation for your valuable advices.

  • The structure of the abstract has been adjusted in the marked revised manuscript according to your advice. (Line 10-36)
  • According to your advice, a hypothesis has been put forward at the end of the Introduction and was verified later in the marked revised manuscript. (Line 82-84)
  • The formats of the statistical calculations in the research methodology have been adjusted in the marked revised manuscript. (Line 123, 147)
  • Table 1 has been adjusted according to your advice, and the necessary information has been added in the legend of Table 1 in the marked revised manuscript. (Line 277-279)
  • The correlations between the obtained phenotypic, physiological and biochemical, and transcriptome results, and the verification of the hypothesis in this study have been added in the Discussion of the marked revised manuscript. (Line 460-475)
  • The Conclusions has been added after the Discussion of the marked revised manuscript. (Line 478-498)

Reviewer 3 Report

The manuscript is interesting. Title is consistent with the content of work. The significance of this study was justified considering the number of literature. Abstract manner contains the necessary information. Tables and figures are informative. In my opinion, the entire work was constructed logically and the study contains a large amount of data, along with good discussions of the results. The authors have provided additional data, descriptions and explanations in supplementary materials.

Some remarks

·  Write a separate chapter on Conclusions in which you briefly summarize your results, and write about the limitations of your research and guidelines for future research.

· Collect all the abbreviations you used in the manuscript in one place. This will make the manuscript easier to read.

·  Improve quality Fig. 4. Now it is hard to read.

·  Make sure all items are cited in the text and in the table of contents, and vice versa.

Author Response

First of all, we would like to express our appreciation for your valuable advices.

  • The Conclusions has been added according to your advice after the Discussion of the marked revised manuscript. (Line 478-498)
  • The abbreviations that used in this manuscript have been collected in “Abbreviations” in the marked revised manuscript. (Line 501-512)
  • We are sorry that the quality of Figure 4 was not very well displayed in the docx. manuscript. The original .eps file of Figure 4 has been uploaded to the submission system, and the distinguishability of the .eps file were higher than 600 dpi.
  • The references have been adjusted according to your advice in the marked revised manuscript. (Line 539-671)